# Hierarchical AI enables global interpretation of culture plates in the era of digital microbiology

Alberto Signoroni [1,2,6] ✉, Alessandro Ferrari[3,4,6], Stefano Lombardi [1,3], Mattia Savardi [1,2], Stefania Fontana [3] & Karissa Culbreath[5]

Full Laboratory Automation is revolutionizing work habits in an increasing number of clinical microbiology facilities worldwide, generating huge streams of digital images for interpretation. Contextually, deep learning architectures are leading to paradigm shifts in the way computers can assist with difficult visual interpretation tasks in several domains. At the crossroads of these epochal trends, we present a system able to tackle a core task in clinical microbiology, namely the global interpretation of diagnostic bacterial culture plates, including presumptive pathogen identification. This is achieved by decomposing the problem into a hierarchy of complex subtasks and addressing them with a multi-network architecture we call *DeepColony*. Working on a large stream of clinical data and a complete set of 32 pathogens, the proposed system is capable of effectively assist plate interpretation with a surprising degree of accuracy in the widespread and demanding framework of Urinary Tract Infections. Moreover, thanks to the rich species-related generated information, *DeepColony* can be used for developing trustworthy clinical decision support services in laboratory automation ecosystems from local to global scale.

Microbiology is faced with tremendous questions, with bacterial, viral, and parasitic infections representing major threats to human health[1]: new species discovered annually[2], re-emerging pathogens[3,4], zoonotic infections[5] and antimicrobial resistance[6]. Correct and timely identification of pathogens is essential to effectively fight infections and the interpretation of bacterial cultures from human collected samples is a pivotal step in the clinical process. Over the last 30 years, molecular biology-based techniques (from PCR to Next Generation Sequencing[7,8]), vibrational spectroscopy[9], and mass spectrometry diagnostic tools such as MALDI-ToF[10,11] have also emerged as critical tools in the clinical microbiology laboratory for identification of pathogens. However, routine bacterial culture continues to be the mainstay for diagnosis of bacterial infectious diseases[12]. The continued use of culture in the clinical microbiology laboratory (CML)[13] is due to its role in the recovery of viable organisms, the availability of a given organism for antimicrobial susceptibility testing (AST), the detection of unusual or unexpected pathogens, and the lower costs associated with culture-based methods compared with culture-independent methods.

Notwithstanding the benefits, culture interpretation is often a challenging undertaking even for the skilled microbiologist. Other specialties in clinical laboratory medicine that require subtle interpretation and complex visual diagnostic tasks have already shifted into a digital imaging ecosystem (e.g., urinalysis, haematology and cytology). However, bacterial culture and culture plate interpretation have remained mostly unchanged from its origins in the 19th century.

[1]Department of Information Engineering, University of Brescia, Brescia, Italy. [2]Department of Medical and Surgical specialties Radiological Sciences and Public Health, University of Brescia, Brescia, Italy. [3]Copan WASP, Brescia, Italy. [4]NVIDIA, Munich, Germany. [5]Department of Infectious Disease, Tricore Laboratories, Albuquerque, New Mexico, USA. [6]These authors contributed equally: Alberto Signoroni, Alessandro Ferrari. ✉e-mail: alberto.signoroni@unibs.it

Thus, the entire culture process is still accompanied by the susceptibility to human factors and is still restricted by staff shift-availability, while the skilled workforce available to perform and read the cultures has decreased in the past few years[14]. In this resource-critical context, the recent advent and deployment of Full Laboratory Automation (FLA) systems[15–17] led to levels of automation and standardization of the physical culture steps (specimen processing, plate streaking and incubation) along with a step of shooting the culture plates, which makes the plate images available for the application of digital analysis tools (Fig. 1a). FLAs have already demonstrated clinical improvements through early growth detection, better culture quality, shorter turn-around times, and significant cost savings, with highly positive impacts on complementary downstream tasks (MALDI-ToF identification, AST, genome sequencing)[18–21]. Despite this, considerable challenges still remain open within the digitized laboratory workflow[22,23] regarding the interpretation of culture images. Image analysis solutions in bacterial culture have been developed in specific contexts[18,24–27] and especially for the use of selective and differential media including chromogenic agar and other differential media. However, algorithms developed using these methods already benefit from the biochemical properties of media and do not require broad identification of the many types of organisms seen in culture. Moreover, to implement such a tool, laboratories would be required to adopt a new media for their culture process, potentially incurring additional costs. Few studies have demonstrated the development of an image analysis algorithm using a non-selective, non-differential media commonly used in the laboratory, such as sheep blood agar. Additionally, consistent and replicable culture interpretation remains a clinical challenge requiring significant expertise[28]. Therefore, there is a need for an additional clinical decision support tool that informs the technologist of the expected action based on observation of morphologies of grown organisms to ensure accurate and consistent culture interpretation on a large scale. Such an automated whole-plate interpretation has still not been tackled in its full complexity, and nothing even close to it has been prefigured until now[16,17,23,29].

We approach these challenging objectives by leveraging the recent digital revolution of clinical microbiology culture and the strengths of deep learning (DL) for the solution of complex tasks[30–33]. In particular, compared to the most common solutions based on single

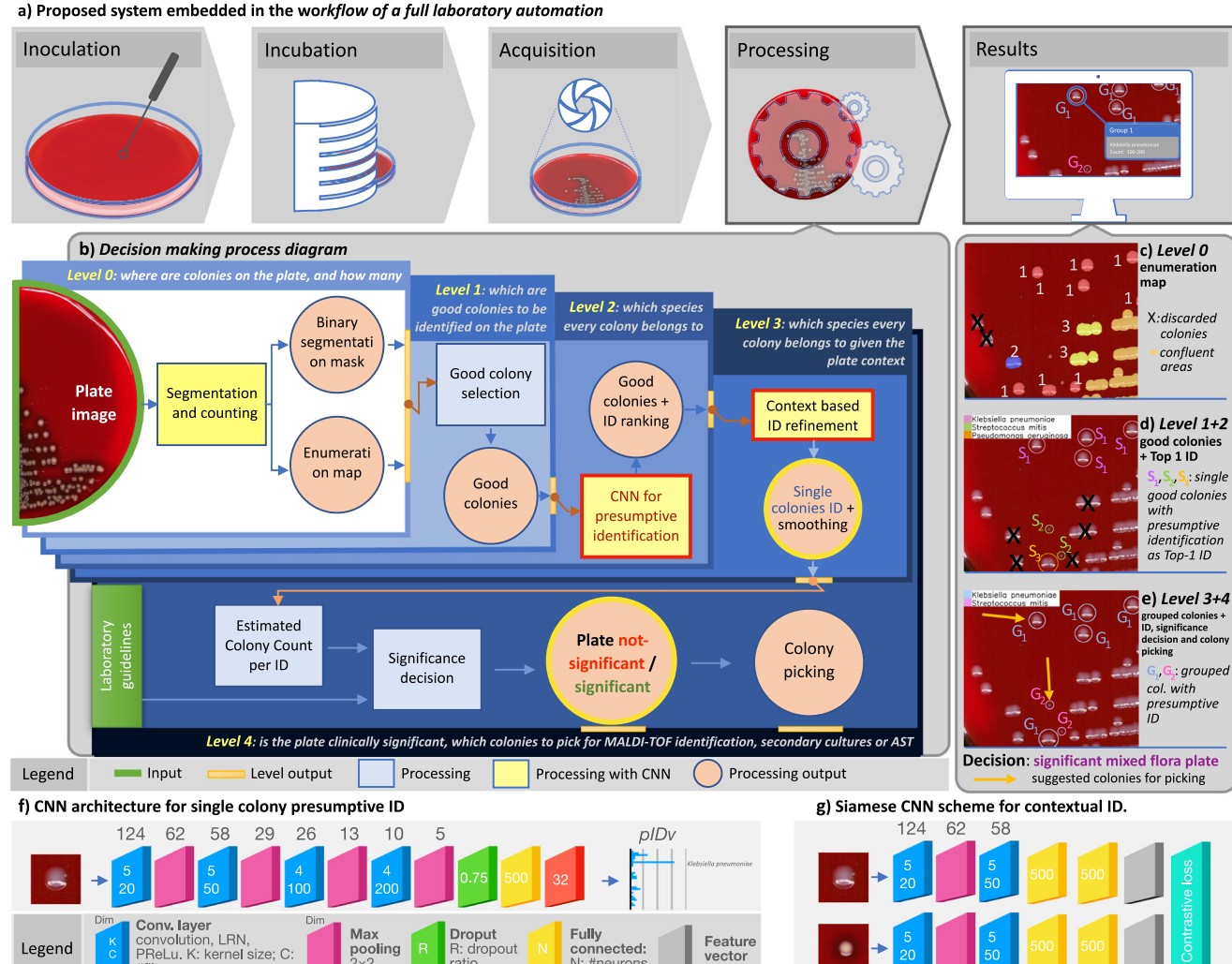

**Fig. 1 | Overview of the *DeepColony* system. a** The FLA workflow and the role of the proposed solutions. **b** *DeepColony* hierarchical architecture of the operational and decision-making processes. **c** Example of "enumeration map" (*level 0* output) where the cardinality of colony aggregates is indicated with colours and numbers. An "X" appears on colonies discarded because they are located in colour distortion areas. **d** Example (on the same plate) of selected "good colonies" (*level 1* output) with their presumptive ID (*level 2* output). An "X" appears on colonies discarded because they are not single or too close to the plate border or to confluence areas. Detected species $S_i$ are indicated for each colony with colours and corresponding legend. **e** Outputs of *level 3* and *4* are shown on the same plate. $G_i$ refers to species ID updated after similarity-driven contextual grouping, while subsequent clinical relevance interpretation indicates a significant mixed flora plate, where best colonies eligible for picking are indicated with arrows. **f** CNN architecture for single colony presumptive ID (red box at *level 2*). **g** "Siamese CNN" scheme (red box at *level 3*) for contextual ID.

convolutional neural networks (CNN)[34], multi-network architectures are attractive in our case because of their ability to fit into contexts where decision-making processes are stratified into a complex structure[35]. The system must be designed to generate useful and easily interpretable information and to support expert decisions according to safety-by-design and human-in-the-loop policies, aiming at achieving cost-effectiveness and skill-empowerment respectively. This requires an advanced ability to combine species-level identification and quantitation across a possibly wide range of clinically relevant pathogens, a skilful combination of multiple visual, cognitive, and procedural tasks, and a computational architecture capable of reproducing this complex environment involving both single-colony and whole-plate analyses.

## Results

### DeepColony architecture

We developed *DeepColony*, a hierarchical multi-network capable of handling all identification, quantitation and interpretation stages, from the single colony to the whole plate level, whose nested architecture is depicted in Fig. 1b (milestones are indicated with green circles) and detailed in "Methods" (DeepColony architecture). Firstly, in the challenging context of urinary tract infections UTI[36], the bacterial species of isolated colonies are identified with a degree of accuracy approaching intrinsic visual discrimination limits for the majority of clinically relevant pathogens. Species identification stems from a dual process: a first "pathogen aware - similarity agnostic" approach focusing on single colony recognition (Fig. 1b, *levels 0, 1* and *2*, Fig. 1c, d), followed by a "similarity aware - pathogen agnostic" refinement step involving the global plate context (Fig. 1b, *level 3*, Fig. 1e). As a second milestone, the possibility to integrate the rich body of species-specific structured information generated from *levels 0-3* into a rule-based decision system is demonstrated, with the aim of providing clinically accurate culture interpretations (Fig. 1b, *level 4*, Fig. 1e). Coherently to its action, *DeepColony* also produces workable information to guide best-colony picking (see Fig. 1e) which is essential for both the confirmation (e.g., MALDI-ToF identification) and the therapy targeting (e.g., antimicrobial susceptibility tests, AST) phases.

### Colony-level dataset and phylogenetic grouping

A large labelled clinical dataset has been created for the training of *DeepColony* at the colony level. Starting from 1351 unique plate images, a dataset of 26213 *isolated colony* images was produced. These represent 32 UTI bacterial and fungal species, constituting 98% of the species that have been observed in three months of the clinical routine of a large CML, and are represented in their clinical variability. To avoid mislabelling, colonies were only drawn from pure flora cultures after MALDI-ToF identification to be used as *ground truth-identification* (GT-ID). Figure 2 depicts four examples for individual species, with the inclusion of polymorphisms and morphologic diversity, along with histograms indicating the number of available colonies in the testing subset of the dataset. All details about the acquisition and composition of this dataset are given in "Methods" (Datasets). Phylogenetic relations between UTI microorganisms have been further considered aiming to arrange the 32 species into main groups. This is highlighted in the central part of Fig. 2, where a subdivision determined by the terminal branches of the diagram leads to 16 groups of clinical relevance.

### Single colony identification (level 0 to 2)

Quantitation of bacterial growth is the first step in the plate culture significance assessment. At *level 0*, in Fig. 1b, a DL-based colony counting method[25] is used to produce an "enumeration map" (depicted in Fig. 1c). At *level 1*, this map is used to select "good colonies", i.e., colonies isolated from the confluent groupings and well developed among all single colonies on the plate (Fig. 1d). Importantly, *level 1* takes into account species polymorphism and colony maturity when

selecting the more reliable colonies. Following the selection of isolated colonies, *level 2* completes the task of attributing a presumptive species-level identification to each selected bacterial colony. To this end, a CNN architecture operating on colony image segments has been designed to generate a "presumptive identification vector" (*pIDv*) consisting of a confidence-based ranking of most probable bacterial species (out of the 32 possible ones) for each image segment generated by *level 1*. The CNN scheme (with an example of the resulting *pIDv*) is shown in Fig. 1f, while technical details and the training using our *colony-level dataset* are described in "Methods" (Single colony identification). At the completion of *levels 0-2* (Fig. 1d) *single good colonies* are identified for a "pathogen-aware, similarity agnostic" description based on the most probable (Top-1 ID) species in *pIDv* for each colony.

Algorithm performance on the testing subset is presented in Fig. 3a by means of a $32 \times 32$ confusion matrix representing Top-1 ID (rows) *versus* GT-ID (columns), in absolute occurrence values, with an overall identification accuracy at this stage of 83.4% compared to the ground truth. Analysing the errors (off-diagonal entries), the main evidence is that most misclassifications follow a block-diagonal pattern. Given the non-casual ordering of species, this is evidence that errors mainly occur within the same phylogenetic group. This is confirmed by assessing the performance on the diagnostic/phylogenetic subdivision of 16 classes (Fig. 3b), where it can be observed that the overall accuracy increases to 88.4%. Given the objective of an assisted presumptive diagnosis and to further exploit the classification confidence information contained in the *pIDv*, it is of interest to include in our evaluation also Top-2 and Top-3 species accuracies, where Top-*n* accuracy is computed by considering samples as 'correct' if the true species is listed in the *n* most probable classes (out of the possible 32) suggested by the network. The Top-2 accuracy is 92.3% while for Top-3 a value of 95.5% is reached. With morphologically similar organisms grouped, most of the residual misclassifications in Fig. 3b remain near the diagonal. This is a further evidence of the added value of the proposed diagnostic/phylogenetic grouping/ordering. For example, small colonies such as *Lactobacillus* species are sometimes confused with *Aerococcus urinae*, and vice-versa; *Streptococcus agalactiae* colonies are sometimes confused with *Enterococcus faecalis* or *faecium* and vice-versa. Similarly, few *Streptococcus mitis* colonies are misclassified as *Pseudomonas aeruginosa*, while some *Enterococcus faecium* and *Candida* species are labelled as *Lactobacillus* species. This morphological misclassification is similar to what may be observed by a technologist. Additionally, the naturally skewed composition of the *colony-level dataset* (only partially compensated for least represented classes) has a slight influence on the balance of the confusion between species. This is observable for *Escherichia coli* (constituting 16% of the dataset), which is more frequently erroneously identified than misclassified as another species. In summary, as far as *level 2*, by combining different interpretative criteria (diagnostic-phylogenetic interpretation and ID vector confidence exploitation), we are able to provide a first classification of organisms into very specific and clinically significant morphologic groups but, interestingly, we have not yet reached all what can be done to support microbiologists toward plate interpretation.

### Context-based Identification (level 3)

In *levels 0-2* each single colony is assessed independently. However, in clinical application, the entire context of the plate must be assessed to determine if observed colonies are similar (pure culture) or different (mixed cultures). At *level 3* a "similarity aware - pathogen agnostic" refinement step is applied, involving the global context of the plate and requiring the computation of a similarity metric. This requires an abstraction from the single colony perspective; now, visually similar colonies, which could possibly belong to different species, must be simultaneously considered and compared to the overall growth, with the aim of improving the consistency of the presumptive identification and of honing the precision of the tool in doubtful situations. This

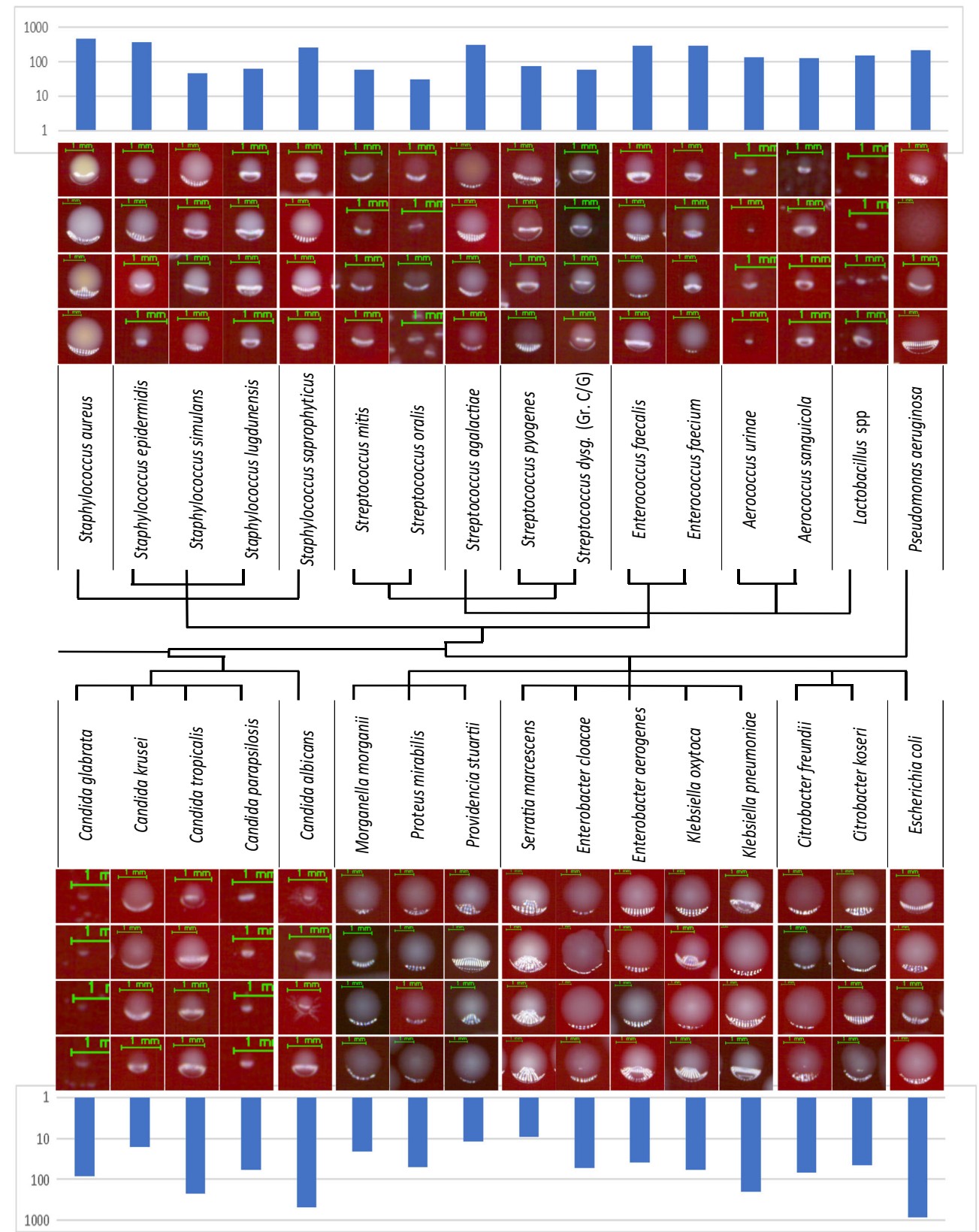

**Fig. 2 | Examples from the 32 pathogen species in the *colony-level dataset*.** Four images for each species are presented to exemplify commonly encountered morphological and spectral inter-species similarity and intra-strain polymorphism. Histograms indicate the number of available colony images in the dataset for each pathogen. The central diagram indicates phylogenetic links and proposed groupings (also delimited by the vertical lines between the pathogen names). Source data are provided as a Source Data file.

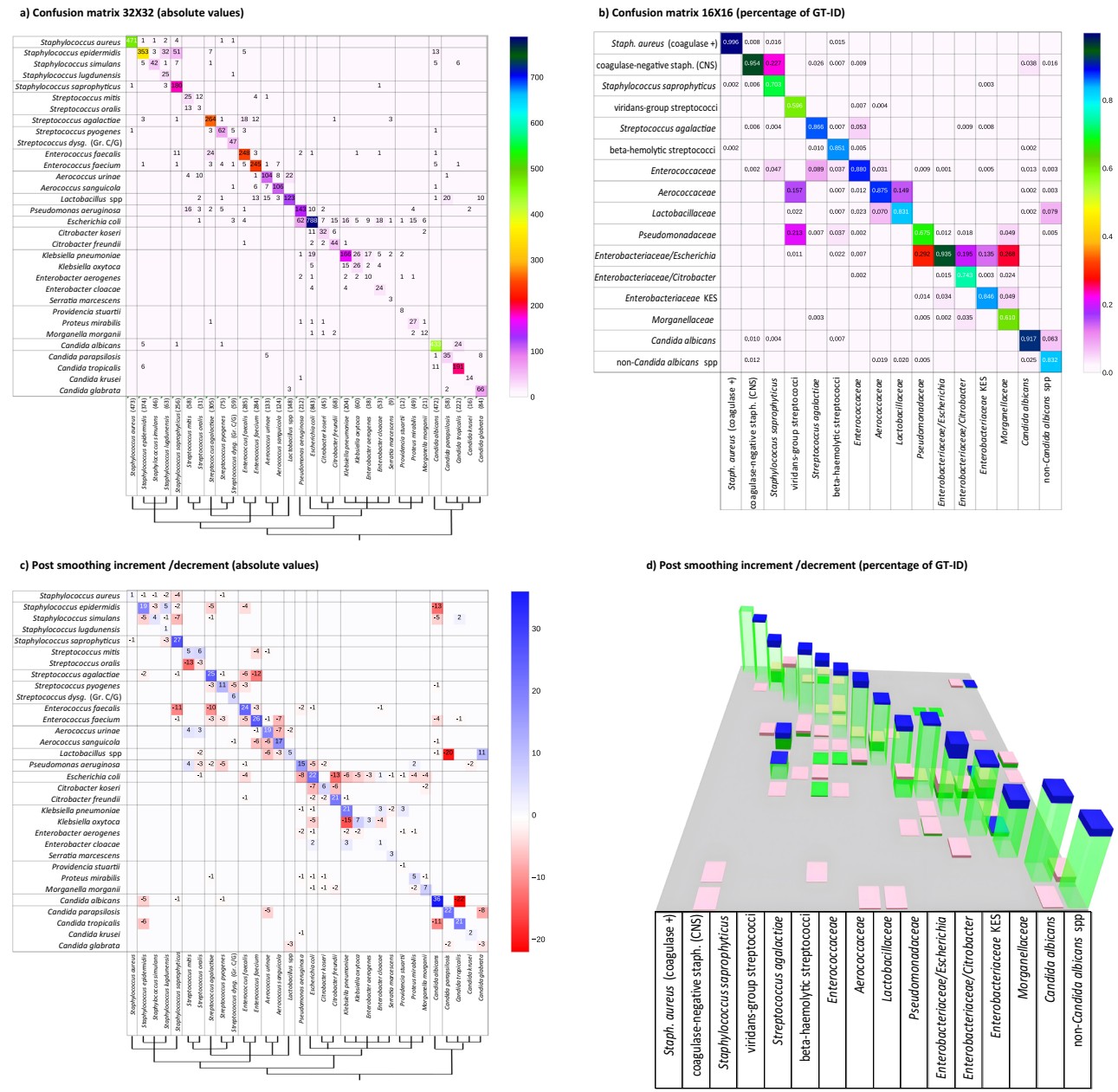

**Fig. 3 | Two-step presumptive identification at the colony-level. a** 32 × 32 confusion matrix for single colony species ID (in absolute value). Phylogenetic group subdivisions are indicated with black lines. **b** 16 ×16 confusion matrix for single colony phylogenic group ID (in relative value referred to the group-based GT).

**c** 32 × 32 matrix indicating relative increase/decrease (in absolute value and referred to **a**) after similarity-based smoothing. **d** 3D 16 × 16 matrix with column height indicating relative increment (blue)/decrement (pink) referred to baseline (green) matrix **b**. Source data are provided as a Source Data file.

delicate task is addressed through a "context-based regularization" approach based on a *non-linear similarity-driven embedding* followed by a *clustering in the embedded space* that enables an enforcement of the identification consistency. For the dimensionality reduction embedding a Siamese CNN[37,38] has been trained whose architecture is shown in Fig. 1g, exploiting an extensive training set comprising 200,000 image pairs (plus 10,000 image pairs for validation) created specifically for this purpose. Once the isolated colonies are mapped in an embedding space, Mean-shift clustering[39] is used to define similarity-based colony clusters. Additional details are discussed in "Methods" (Context-based Identification). A final classification is given for each identified cluster by averaging the *pIDv*'s from *level 2* within the cluster, and this allows an increase in classification consistency. As shown in Fig. 1e, the ID labels are now assigned to group of colonies instead of single colonies (and accordingly the *pIDv*'s). This

regularization method resulted effective in improving overall identification performance and interpretation accuracy due to the quality and coherence of the embedding and clustering combination. On the 32 bacterial species, Top-1 accuracy increased to 90.6% (Top-2 96.6%, Top-3 97.6%) which is an increment of 7.2 percentage points (4.3 and 2.1 pps for Top-2 and Top-3 respectively). Figure 3c reports absolute values of increased (blue) or decreased (red) accuracy related to each species, compared to the *level 2* values in Fig. 3a. Consistent with the significantly increased accuracy, blue squares are mainly localized on the matrix diagonal, while out of diagonal red squares show significant misclassification reductions. A 3D representation of variations in the 16×16 confusion matrix of the diagnostic/phylogenetic species aggregations is shown in Fig. 3d, where green columns correspond to Fig. 3c values, and the blue/pink bars indicate the increment/reduction portions. Improved accuracy (i.e., diagonal increments and out of

diagonal reductions) is observed in the large majority of cases, with an overall accuracy of 93.8% (5.4 pps increase). As a mild side effect, the sum of misclassification of both *Streptococcus mitis* and *oralis*, instead of balancing between them, accounts for the slight increment in misclassification of viridans-group streptococci as *Aerococcaceae* and *Pseudomonadacee* (+7.9% and a +1.1% respectively) while the correct classification decreases (pink top on green column representing −5.6%).

In *level 3*, we thus demonstrate that applying a "similarity-aware, pathogen agnostic" refinement improves the specificity of the organism identification, particularly in the context of mixed cultures.

### Decision support to Culture Interpretation (*level 4*)

Individual colony interpretation and contextual interpretation provides information for the laboratory to use. However, the laboratory must ultimately synthesize that information to generate a laboratory result based on culture type and laboratory-specific rules. In *DeepColony* this is achieved at *level 4*, where information from *level 0* (segmentation and enumeration) and *level 3* (*pIDv*) are combined to compute min-max colony counting range for each identified species on the plate. This opens up the possibility of assisting the decision-making process that is codified in the CML guidelines for the interpretation and subsequent management of culture plates.

A pictorial example of decision about plate significance is given in Fig. 1e, where the plate is interpreted as "mixed flora culture with two significant species". One colony for each species is also indicated for possible colony picking, which is crucial for the following ID confirmation (by MALDI-ToF) and/or therapeutic assessment (by AST) phases. More in detail, the whole *DeepColony* decisional process is illustrated in Fig. 4 on three representative plates, exemplifying all the different possible outcomes (in presence of bacterial growth): significant pure flora plate, significant mixed flora plate, non-significant mixed flora plate. *Level 0* and *level 1* steps are represented for Plate1

only (Fig. 4a–d), while *level 2-4* (Fig. 4e–o) are exemplified on all the three plates. For *level 2* plate images, single good colonies are marked with coloured circles, indicating the Top-1 ID species of *pIDv* vector associated to each colony. For *level 3*, coloured circles now indicate the Top-1 ID species of *pIDv* vector assigned to grouped colonies after context-based regularization. For Plate2 only, the presumptive identification scores related to Top-1 ID, Top-2 ID and Top-3 ID of the *pIDv* vector are shown for each identified species at *level 2* and after group-based contextual regularization at *level 3*. Plate-level interpretation is generated at *level 4* combining the produced species identification and the other quantitative info with the CML rules that, for our case, are depicted in Fig. 5a and detailed in "Methods" (Culture Plate Interpretation). For example, the presumptive presence of only two different species on Plate2 (according Top-1 ID of each group), together with CML rules, determine a "significant mixed flora" interpretation with the indication of the identified infective species. This comes with the generation of associated values e.g., in terms of presumptive identification scores Top-1 ID, Top-2 ID and Top-3 ID (coherently with the first three confidence values of the *pIDv* vector) for each good colony and for the grouped species, as detailed in Table 1. Similarly, Plate1 is interpreted as "significant pure flora", while Plate3 is interpreted "non-significant mixed flora". An additional panel of 9 plates, representative of other culture scenarios is provided in Supplementary Fig. 1. The variegated visual and quantitative information generated constitutes a transparent and informative explainability pattern allowing the technicians to well interpret the various phases of the machine-driven decisional processes.

### Plate-level decision (*DeepColony* vs humans) on a large clinical dataset

Given the ability to perform these complex interpretations, *DeepColony* has been applied to a large clinical dataset of 5,051 urine cultures acquired in a large US laboratory. The *plate-level clinical dataset*, more

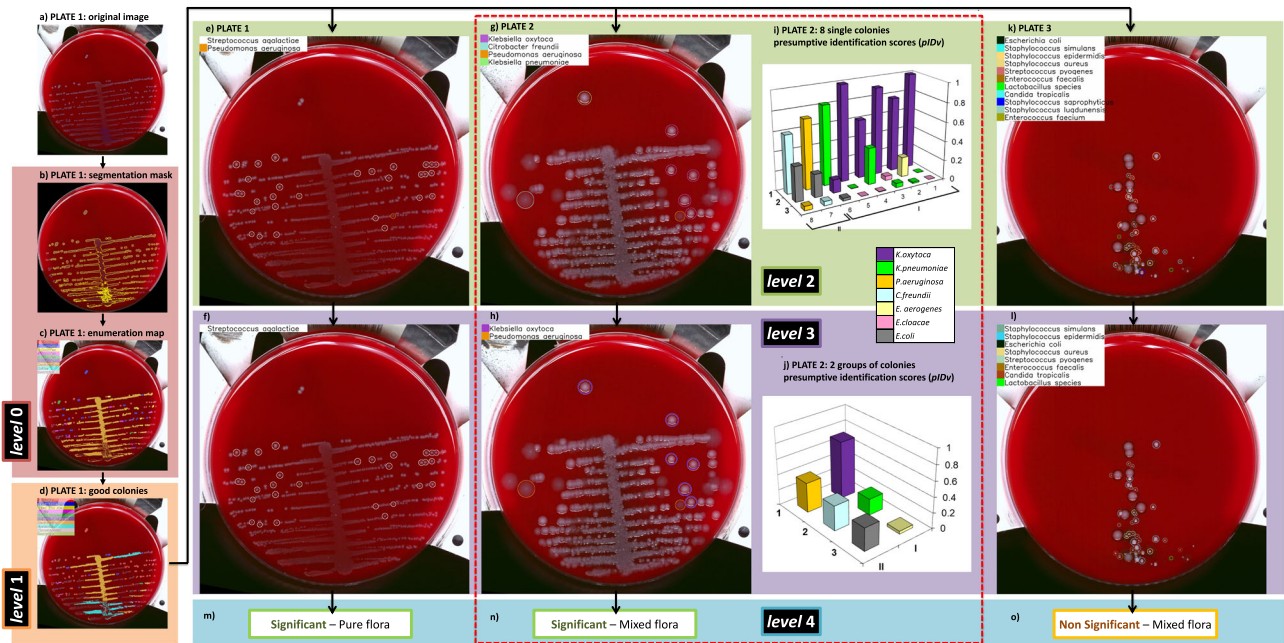

**Fig. 4 | *DeepColony* in action on three sample clinical plates (P1, P2 and P3). a** Original P1; **b** P1 segmentation mask (*level 0*); **c** P1 enumeration map (*level 0*); **d** Good colonies on P1 (*level 1*); **e, g, k** *level 2* Top-1 ID on P1, P2 and P3 respectively, where different colours are used to mark single good colonies according to the identified species as listed in the plate legends (*Streptococcus agalactiae* and *Pseudomonas aeruginosa* for Plate1 in **e**, *Klebsiella oxytoca, Citrobacter freundii, Pseudomonas aeruginosa* or *Klebsiella pneumoniae* for P2 in **g**, eleven different species for P3 in **k**).; **f, h, l** *level 3* contextual Top-1 ID on P1, P2 and P3 (*Streptococcus*

*agalactiae* for Plate1 in **f**, *Klebsiella oxytoca* and *Pseudomonas aeruginosa* for P2 in **h**, eight species for P3 in **l**); **i** 3D representation of the first 3 entries (Top-1 ID, Top-2 ID and Top-3 ID) of the *level 2 pIDv* for each good colony of P2 (different colours indicate different species) and anticipated indication of following grouping I and II; **j** the same 3D representation for grouped species operated in *level 3*; **m, n, o** Plate-level interpretation generated at *level 4* by the combination of species-related information (ID and quantitation) and CML rules (for this case represented in the block diagram of Fig. 5a). Source data for **i** and **j** are provided as a Source Data file.

**Fig. 5 | *DeepColony* in the laboratory routine. a** Shared CML rules for plate significance attribution (here NGUF stands for Normal genitourinary flora). **b** *DeepColony* vs Laboratory performance **c** Laboratory Impact of machine-assisted plate interpretation.

**Table 1 | Values of *pIDv* for each good colony and for grouped species referred to PLATE2 of Fig. 4**

| Colony n. | Top-1 ID | *pIDv* value | Top-2 ID | *pIDv* value | Top-3 ID | *pIDv* value |
|---|---|---|---|---|---|---|
| 1 | *K.oxytoca* | 0.9991 | *K.pneumoniae* | 0.0008 | *E.cloacae* | 0.0001 |
| 2 | *K.oxytoca* | 0.7754 | *E. aerogenes* | 0.2067 | *K.pneumoniae* | 0.0126 |
| 3 | *K.oxytoca* | 0.9111 | *E.cloacae* | 0.0446 | *K.pneumoniae* | 0.0439 |
| 4 | *K.oxytoca* | 0.6104 | *K.pneumoniae* | 0.38098 | *E.cloacae* | 0.0042 |
| 5 | *K.oxytoca* | 0.9974 | *K.pneumoniae* | 0.0024 | *E.cloacae* | 0.0001 |
| 6 | *K.pneumoniae* | 0.8237 | *K.oxytoca* | 0.1356 | *E.coli* | 0.0274 |
| 7 | *P.aeruginosa* | 0.7192 | *E.coli* | 0.2355 | *C.freundii* | 0.0427 |
| 8 | *C.freundii* | 0.5966 | *E.coli* | 0.355 | *P.aeruginosa* | 0.0435 |
| **Group of colonies n.** | **Top-1 ID** | **pIDv value** | **Top-2 ID** | **pIDv value** | **Top-3 ID** | **pIDv value** |
| I | *K.oxytoca* | 0.7382 | *K.pneumoniae* | 0.2107 | *E. aerogenes* | 0.0357 |
| II | *P.aeruginosa* | 0.3813 | *C.freundii* | 0.3196 | *E.coli* | 0.2953 |

Confidence values (presumptive identification scores) are given on a 0 to 1 scale.

details in "Methods" (Datasets), was acquired along with the technologists' interpretation in terms of *no growth, contaminated (mixed)* or *positive* culture, according to the laboratory reading guidelines (the same depicted in Fig. 5a and adopted for *DeepColony*-guided interpretation at *level 4*). In Fig. 5b, *DeepColony* interpretation is compared with that of the technologists on the entire set of cultures to determine both class-based and overall human-machine agreement. Despite the intrinsic task difficulty, a high overall agreement of 95.4% (unweighted Kappa of 0.920) between machine-based and manual presumptive identifications is reached[40], with a class-based peak of 99.2% for the negative "no-growth" cultures, and a high agreement of 95.6% also reached for positive cultures. Out-of-diagonal values in the confusion matrix of Fig. 5b remain noticeably low and can be considered marginal, with the exception of the 194 plates, the 7.4% of the 2623 cultures, interpreted as positive by *DeepColony*, that were instead manually interpreted as contaminated. This lead to a lower agreement of 77.1% for the mixed growth or "contaminated" cultures due to a deliberately precautionary behaviour, which is related to "safety by design" criteria: the upper bound of the estimated number of colonies identified within a species is considered, while additional qualitative considerations are limited to allow for technologist interpretation (who may deem a plate as non-significant despite some above-threshold CFU counting) aiming at reducing the risk of improperly discarding potentially positive cultures.

Taken together, this study shows that *DeepColony* can effectively classify and enumerate organisms to be combined with laboratory-based rule systems to augment the microbiologists' clinical decisions on the interpretation of clinically significant cultures.

## Discussion

Human-driven interpretation of culture plates requires the presumptive identification and quantitation of clinically relevant pathogens at both the colony- and plate-level. This reciprocal and dual perspective drove the design of the *DeepColony* architecture (Fig. 1) aimed at assisting such a challenging visual task and consequent

analytical steps. In *levels 0-2* each individual colony is assessed for identification while at *level 3* a contextual refinement of each colony in relation to other colonies is performed. Experienced microbiologists consider the relative variation of colony dimension, position and maturation to identify (or exclude) polymorphism and thus reliably classify colonies. Trained on enough plates, our system contextually learns both these similarities and differences within a species, in order to correctly interpret not only inter-species similarity but also intra-species or better, since referred to the single plate, intra-strain polymorphism. This multi-level AI-driven action enables species-specific identification and quantitation, and this is unprecedented in a machine-assisted context. The rich body of generated structured information is then used to feed a decision support system compliant to the diagnostic procedures adopted within the CML (*level 4*) to provide an overall interpretation of the culture.

The results of this study demonstrate a high level of agreement between *DeepColony* and the interpretation given by the technologist, in particular for the negative cultures (>99% agreement). Among the positive cultures, *DeepColony* tended to interpret as positive a portion of cultures that were interpreted as contaminated by the laboratory. This is however due to safety-by-design (false negative culling) settings, since the system is here designed as a screening tool to correctly call true negatives allowing some false positive results. This still provides an advantage in which repetitive work reviewing negative results is lightened, and professionals are better able to focus on the most relevant and/or critical cases. It also enables a series of improvements to the diagnostic process (e.g., traceability, higher degree of standardization, prioritized scheduling), which could result in increased personnel skill and rewards. Ultimately, by giving fully informative decision support and reliably speeding up presumptive diagnosis, a machine-assisted plate interpretation creates the conditions for significant positive repercussions, in terms of precision and timeliness, on the whole infection management workflow, with the prospect of improved and highly responsive patient infection care on a large scale.

Moreover, it is increasingly clear that data-driven AI systems in critical sectors must not only have demonstrable technical functionalities but, to be of real support and to overcome the demands of the emerging regulations[41], they must be designed to guarantee trustworthiness thanks to adequate levels of transparency and explainability[42]. Considering the above observations, our work fits well in the context of liability and can be included in a broader framework for the development of assistive tools for FLA empowerment. This global picture is summarized in Fig. 5c where the impact multiplication factors are evidenced and linked to the main traits of the proposed approach. *DeepColony* can be therefore configured to safely reduce the laboratory workload and to make decisions which are highly coherent with assigned interpretation guidelines, guiding and supporting the specialists in focusing on more relevant case reports with an increased level of information and assistance to better set the downstream steps.

An intrinsic limitation of *DeepColony* is the current inability to reliably identify species inside confluent areas. This forced us to introduce prudential ranges (i.e., confidence intervals) in the final phase of species-level concentration estimation. However, if there are multiple organism morphologies, the culture may be classified as "contaminated" or flagged for additional review. Because of the safety-by-design feature intrinsic to this system, this does not appear to be to the detriment of the consistency of the results. The substantial improvement we observed in *pIDv*s' estimations led by the contextualized reasoning (which grounds on already reliable single-colony identifications) is further evidence of the fact that residual difficulties encountered by our system are coherent with the actual limits of expert visual interpretation: colonies from distinct species that are indistinguishable or, conversely, high polymorphic variability that risks inducing undesired differentiation within the same species. Additionally, the system can take full advantage of the high quality and resolution of the FLA digitized images, well over what is visible to the naked eye in traditional CML activities. Eventually, in this work we evaluated urine cultures that represent a large variety of organisms. However, there are other organisms that were not evaluated in this analysis that may be prevalent in other culture types (e.g., Streptococcus pneumoniae in sputum, Neisseria meningitidis in CSF, Neisseria gonorrhea in genital). In these cases, the models would need to be adapted to reflect the range of organisms seen in a particular culture or region.

In conclusion, *DeepColony* is a unique framework for improving the efficiency and quality of massive routine activities and high-volume decisional procedures in a microbiological laboratory, with great potential to refine and reinforce the critical role of the microbiologist.

## Methods

### Datasets
The image datasets utilized in this study comprise both colony-level and plate-level data. These images come from high-resolution digital scans of cultured plates acquired from clinical specimens processed by WaspLab™ (by Copan WASP®, Italy) FLA facilities. Digital images were produced using the WaspLab™ 16Mpixel tri-linear colour camera performing pushbroom line-scanning under a calibrated white LED lighting system. Combined with high-quality telecentric optics, this guarantees low geometric distortion and high spatial resolution (24.5 μm/pixel).

**Colony-level dataset.** This clinical dataset was generated from plates cultured in collaborating labs in the US, where colony images were collected in a fully anonymised form, without any kind of interaction or interference with the laboratory and patient diagnostic/prognostic/therapeutic processes. Standard protocols for specimen processing, plate incubation and digital image generation were followed[19-22]. Specifically, all plates, except for a small number of cultures derived from ATCC strains, were inoculated with 1 μl loop on REMEL™ blood agar

from urine samples non-invasively collected. Plate incubation occurred in a $CO_2$ atmosphere at 37 °C, and digital images of all plates were captured between 17 and 23 hours of incubation. Ground truth identification came from Vitek™ MALDI-ToF laboratory analyses. The size of colonies varies widely, ranging from approximately 0.4 mm (~20 pixels) to about 7 mm (~300 pixels) in diameter. Colony-level (i.e., single colony) images were provided using *level 0* segmentation (Fig.1a)[25], which creates a mask that excluded confluent areas and colonies with underdeveloped morphology (due to proximity to highly confluent growth or the plate border) as they are not representative of the bacterial colony morphology. Additionally, pixels outside the main colony segment were set to zero to remove clutter (portions of other segments in the neighbouring area). To standardize colony-level data, images containing single colonies were padded with a fixed 15 pixels margin along the longer edge, and a suitable margin on the other edge for making them square.

A quantitative description of the *colony-level dataset* is provided in Supplementary Table 1. From 1351 *pure flora plate* images (1321 from clinical specimens and 30 from ATCC), 26,213 *isolated colony* images (24,781 clinical and 1432 from ATCC) were extracted. A 60/20/20 subdivision in training/validating/testing subsets was performed at the plate level (random selection among each of the 32 represented species) in order to avoid having colonies from the same plate appearing in different portions of the *colony-level dataset*. The table also provides information about the 16 proposed groups, which were derived from phylogenetic relations and diagnostic relevance criteria. In particular, phylogenetically related species, like *Morganellaceae, Aereococcaceae, Enterococcaceae*, were collapsed together, as were pathogens where a joint identification could have a most diagnostical/clinical relevance than separate identifications (e.g., *Enterobacteriaceae* KES, grouping the pathogens *Serratia marcescens* and *Enterobacter cloacae* with *Enterobacter aerogenes, Klebsiella oxytoca* and *Klebsiella pneumoniae*). However, in some cases, despite the phylogenetic relationship with neighbour species, the clinical relevance of a single species led us to maintain it as a separated category, e.g., *Staphylococcus saprophyticus* versus other coagulase-negative staphylococci (CNS), or *Streptococcus agalactiae* versus beta-haemolytic streptococci, or *Candida albicans* versus other non-*Candida albicans* species.

**Plate-level clinical dataset.** comprises 5,051 WaspLab™ clinical images (from same number of plates) taken from the whole flow of about one week of examinations from urine sample cultures performed at TriCore Reference Laboratories in Albuquerque, NM (US). As for the *colony-level dataset*, all the plates were inoculated from urine clinical specimens using a 1 μl loop on REMEL™ blood agar, and incubated from 17 to 23 h in $CO_2$ at 37 °C. After screening them with a Sismex TM urine analyser, which identifies negative urine samples that can be excluded from the culturing process[43], plates were labelled by the staff on duty according to shared laboratory rules (depicted in Fig. 5a and detailed below), nominally adopted by both lab technologists/microbiologists and *DeepColony*, to determine the clinical significance of the culture.

### DeepColony architecture
*DeepColony* is a comprehensive framework for colony identification and analysis conceived to operate in high-throughput microbiology laboratories. It is constituted of different CNNs that exchange information and cooperate in a hierarchical structure. As depicted in Fig. 1b, this structure is organized in 5 levels, from *level 0* to *level 4*, each one addressing a specific aspect of the analysis, allowing to work at different interpretation scales, from the individual colonies to the whole plate. A pictorial representation of the outcomes of the different levels is presented in Fig. 1c–e, while a detailed algorithmic description of the whole process is given in the Supplementary Pseudocode.

At *level 0*, *DeepColony* determines the locations and quantifies the number of colonies present on the plate. The objective of this initial step is to provide essential information about the spatial distribution and abundance of colonies, being able to identify isolated colonies, to recognize and count touching colonies in small aggregates (up to 6), to detect larger confluence areas and to possibly discard outliers (e.g., dirt material) or colonies growing too close to the plate borders. The colony segmentation and enumeration maps are generated by an approach that exploits a first CNN which have been described in a previous publication[25] and will not be detailed here. As can be seen below, these are not only essential for the interpretation of individual growths but also contribute greatly to the interpretation of the entire plate. An example of the intermediate output of *level 0* is shown in Fig. 1c.

At *level 1*, the system identifies isolated colonies that are good candidates for further analysis and identification. This is why, according the same criteria microbiologists use to select colonies for visual interpretation, the selection of "Good colonies" involves discarding colonies located close to large confluent areas or to other colonies, as they may struggle in obtaining proper nutrients for their own growth, and therefore may not have a sufficiently developed morphology indicative of the phenotypes of their species. Similarly, colonies in proximity of the plate border were also excluded, where not already excluded at *level 0*, this time because specular reflections and geometrical obstacles may make them unsuitable for the tasks. Colonies are discarded based on minimum distance criteria and exploiting a distance transform operated on the segmentation mask provided by *level 0*. In Fig. 1d good colonies are all those circled and thus ready for the subsequent identification and interpretation steps. The core part of *DeepColony* is represented by *levels* from *2 to 4*. Their role and main traits are summarized here, while their detailed description is given in the following paragraphs.

At *level 2*, *DeepColony* makes an initial identification using a CNN that assigns each colony to a particular species, based on visual appearance and growth characteristics. In Fig. 1d three bacterial species have been identified on the plate and indicated with differently coloured circles and symbols $S_i$.

On this basis, *level 3* improves the identification robustness and accuracy by incorporating a smoothing of the identification rankings produced by *level 2* on the basis of a contextual analysis of the bacterial growth on the plate under analysis. An embedding of single colonies in a similarity space is performed using the feature vector generated by a branch of a Siamese CNN trained to recognise morphological variants of the same strain growing on a plate as similar (i.e., close to each other in the embedding space). This makes a subsequent clustering in the embedding space a meaningful way to regularize the species identification rankings within each cluster with a significant beneficial effect on the identification performance. A clear effect of this smoothing can be seen in Fig. 1e where the number of identified strains is reduced thanks to the *level 3* identification refinement.

Eventually, at *level 4*, *DeepColony* assesses the clinical significance of the entire plate and can assist in deciding which colonies should be picked for downstream phases, such as MALDI-ToF identification, secondary cultures, or antimicrobial susceptibility testing (AST). This decision-making process takes into account not only the identification results but also adheres to the specific laboratory guidelines to ensure a proper supportive interpretation in the context of use. Figure 1e indicates the final decision taken on the example plate and the suggested colony to pick for each identified species.

### Single colony identification

The CNN for single colony identification schematized in Fig. 1f includes four convolutional layers and one fully connected layer. The hyperparameters of the architecture, including the number of convolutional layers, hidden units, and filter sizes, were carefully tuned using an extensive grid search. Interestingly, the final optimal CNN configuration is substantially equivalent to the best performing topology for cardinality classification also reported for *level 0* analyses[25], thus confirming the adequacy of the designed deep neural network architecture and capacity to the problem at hand. Other tests made to justify and confirm other architectural choices (type of pooling and overall complexity) are reported in Supplementary Table 2. Further architectural details are available in Supplementary Note 1.

### Context-based Identification

Pathogenic species identification using visual data is challenging because of the numerosity of pathogens, the frequently high inter-species visual similarity, but also different forms of intra-strain morphological variability, in both colony dimensions and aspect, that frequently occur even on the same plate (see Supplementary Fig. 2). Hence the need to find a non-trivial similarity assessment strategy capable to both distinguishing visually similar different species and, at the same time, accommodate for different morphotypes of the same species. Even focusing on single colony images, the original data space is high-dimensional and too complex to directly define suitable mappings and/or metrics for assessing such species-related similarity assessment (this has been experimentally verified and reported in Supplementary Fig. 3). To overcome this challenge, a *Nonlinear similarity-driven embedding* was introduced, wherein a data-driven mapping function is employed to convert high-dimensional data to a lower-dimensional space. This is followed by a *Clustering in the embedding space*, where similarity clusters are exploited within a regularization strategy capable to improve, by a significant margin, the overall quality of colony identification, with obvious positive repercussion on quality and reliability of the pre-clinical interpretation of the culture plates. Both these steps are detailed below.

**Nonlinear similarity-driven embedding.** The adopted approach, based on a *Siamese neural network*[37,38], can learn a mapping function from colony images to a reduced-dimensionality space capable of generalizing on unseen data. This architecture, depicted in Fig. 1g, employs two Siamese-CNN subnetworks (S-CNN), which share the same weights while working on two different input images. Using a large training dataset of 200,000 pairs and a validation set of 10,000 pairs generated from the colony identification (*level 2*) dataset, the network was trained to identify genuine and impostor pairs. To ensure good interpretation conditioning of the plate context, same-strain/different-plate pairs were not considered for training purposes. As the goal of this embedding does not involve maximizing a specific accuracy metric, the model selection process was more indirect, with performance evaluated based on the subsequent clustering step. To follow the performance of the model during training, the loss function was monitored on the validation set, expecting it to decrease steadily. The dimension of the embedding space is defined by the last network layer of Fig. 1g and was experimentally set to 15. Further architectural and training details are available in Supplementary Note 2. Once the network has been trained to recognise strain similarities that are expressed on the same plate, one of the twin branches of the Siamese CNN is used to generate a feature vector to embed colonies in a plate-specific similarity space suitable for subsequent ID regularization.

**Clustering (mean-shift) in the embedding space.** To perform similarity-driven clustering in the embedding space a Mean-shift technique[39] is adopted. This method was selected due to its ability to identify clusters with arbitrary shape and its minimal requirements in terms of feature space topology. Additionally, Mean-shift clustering does not require strong a-priori knowledge, which is often necessary for methods like k-means clustering. The computational efficiency of Mean-shift clustering also made it a favourable choice. Gaussian

kernels are adopted with bandwidth experimentally optimized on the dataset. To simulate mixed flora plate occurrences, virtual plates were generated containing up to three different strains on each plate. A total of 1000 virtual plates were created to enable the construction of a realistic experimental setup to train and test our clustering-based approach. This was necessary since complete knowledge of every colony on mixed flora plates by MALDI-ToF identification is not available, and it would be unfeasible and/or not fully justifiable to obtain such information at a large scale. To compare the partition proposed by the clustering algorithm with the ground truth, an overall evaluation indicator *v-measure* is adopted[44] that combines entropy-based scores indicating homogeneity and completeness of the clustering (a deeper explanation regarding these clustering metrics is provided in Supplementary Note 3 while the obtained best parameters configuration is reported in Supplementary Table 3).

## Culture plate interpretation

With information generated from *level 0* to *3*, both lab technicians and *DeepColony* can operate according to the same shared rules adopted by the CML to determine whether the growth on each plate is significant. Plate interpretation for clinical significance (*level 4*) is typically shaped by widely shared guidelines from which laboratories can derive their own specific protocols. The set of rules in use in our reference CML are depicted in Fig. 5a: (1) cultures with fewer than 10 colonies are considered *no significant growth*; (2) plates with 1–2 detected colony morphologies, or with ≥3 different colony morphologies where one pathogen is prevalent (over 50 CFU) over the normal (under 50 CFU) genito-urinary flora (NGUF), are considered *positive*, thus to be further evaluated for possible downstream steps (Maldi-ToF ID/AST); (3) otherwise, cultures are considered *contaminated*—that is, when they manifest ≥ 3 different colony morphologies without a particular pathogen that is prevalent over the others.

## Reporting summary

Further information on research design is available in the Nature Portfolio Reporting Summary linked to this article.

# Data availability

All data supporting the findings described in this manuscript are available in the article and in the Supplementary Information. Source data for all data figures are provided with this paper. The core colony image dataset is available on Figshare with the following identifier (https://doi.org/10.6084/m9.figshare.24203961). The use-case plate dataset is property of Tricore Laboratories (Albuquerque, New Mexico, USA), it was used under license for the current study and is not publicly available. Requests can be addressed to the corresponding author (expected response time 2 weeks) and access will require explicit permission from Tricore Labs. Source data are provided with this paper.

# Code availability

A detailed pseudocode has been provided in the Supplementary Information. The implementation encompasses custom C blocks integrated into Python 2, with models structured in the Caffe framework. The code (architecture and trained models) are not publicly available due to some proprietary restrictions. They can be made available from the authors on request and with permission of Copan WASP.

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

## Acknowledgements

This work was supported by the Italian Ministry of Research (Smart Factory project CTN01_00163_216730, A.S., A.F., S.L.) and by collaborative university-industry research activities with Copan WASP srl, Brescia, Italy. Authors want to thank Mario Savarese, Stefano Oliani, Giorgio Triva and Stefania Triva from Copan for their support in the realization of this work. This work is dedicated to the beloved memory of Daniele Triva.

## Author contributions

A.S. and A.F. contributed equally. A.S. coordinated all the activities and the writing of the manuscript. A.F. led the development of the core technology and first validations. S.L. and M.S. contributed to both the architectural development and the final validation activities. S.F. contributed to the statistical analysis and representation of the experimental data and to the writing of the manuscript. K.C. conducted the urine culture interpretation portion of the study and participated in the writing of the manuscript.

## Competing interests

We declare no competing non-financial interests, but the following competing financial interests: A.S. and M.S. were supported by university-industry research contracts from Copan WASP srl (Brescia, Italy) in applying AI in lab automation imaging, A.F. and S.F. were Copan employees, S.L. is a Copan WASP employee, K.C. has been a clinical consultant for Copan WASP.
