## [Peer Review File · Nature Communications]

Hierarchical AI enables global interpretation of culture plates in the era of digital microbiologyREVIEWER COMMENTS

Reviewer #1 (Remarks to the Author):

In this manuscript, the authors developed a hierarchical multi-network capable of handling identification, quantitation and interpretation stages, from the single colony to the whole plate level, in the challenging context of urinary tract infections.

Working on a large stream of clinical data and a complete set of 32 pathogens, the proposed system proved to be capable of effectively assisting plate interpretation with a surprising degree of accuracy. Thanks to the rich species-related generated information, the authors claim that DeepColony can be used for developing trustworthy clinical decision support services in lab automation ecosystems from the local to the global scale.

Although I do not have specific knowledge on hierarchical AI, deep learning and Convolutional Neural Networks, I read with interest this manuscript from the view point of a clinical microbiologist.

As a general view, the observation that the overall accuracy was high, is very promising.

I have some general comments for the authors consideration:

- Only blood agar plates were used, on which almost all urinary bacteria are grown and the discrimination of colonies is rather difficult. Why the selective Mac Conkey plates were not used at all, particularly when UTIs were investigated, in which mostly Gram-negatives are yielded and the majority of Gram-positive colonies are contaminants? In Mac Conkey plates, other properties, such as lactose fermentation would help the identification.

- The quality of plating in figure 5 is quite low; the growth in two of the plates is rather confluent, the colony discrimination poor and the enumeration very challenging. Although the authors claim that 1- μ l loops were not used, the pictures look more of 10- μ l loop. Anyway, better discrimination and picking up of pure colonies would be needed for such a system to perform appropriately. Development of such strategy would be important for the system in the future.

- I would like to see 1-2 figures of real-life plates from positive urinary cultures (with one or two different and not so many different colony types), so that the reader may estimate the accuracy of colonies' enumeration and reporting.

- The paragraph lines 294-316 are mainly general, introductory comments and not methods.

- It would be of considerable interest if other types of clinical samples, further than UTI ones, were also tested.

Reviewer #2 (Remarks to the Author):

I want to start this review with a clear statement that I have a pure machine learning background with a few papers in automatic microbiology. Moreover, I assume that Nature Communication is not a purely microbiological journal, like Trends in Microbiology, and articles presented in this journal should be directed to a wider audience than only microbiological experts. Therefore, my review should be considered as a review from a potential Nature Communication reader with a machine learning background.

Overall, it is clear to me that the authors have extensive knowledge in microbiology and have published many papers on this topic. However, the article is very unclear for somebody who does not have such an experience. The descriptions are too long and very complicated, and I guess only a few people in the world who are on the same level of expertise as the authors would be able to read this article and get something interesting from it.

Taking this into consideration, I decided to read Intro, Results, and Discussion very fastly and concentrate on the Method section, which I thought should be of interest to somebody like me - i.e. machine learning expert who cooperate with microbiologists.

Unfortunately, the Method section was also unclear to me. Figure 1a (Overview of the DeepColony system) brings more confusion than clarifications. There is not a single figure of network architecture. Reading the paragraph "Single colony identification", I first thought that the authors use simple CNN architecture, but then "Context-based Identification" appeared with "Siamese" networks, and the description is too fuzzy to understand how they are used and why? The authors bring too many implementation details, like PreReLU, momentum, etc, which should be moved to supplementary materials.

In my opinion, the current version of the article should not be accepted, and the editor should ask the authors to rewrite the whole method section using best practices from other articles already published in Nature Communication, like this one: <https://www.nature.com/articles/s41467-019-12898-9>

This already-published article, in contrast to the reviewed one:

- Has a shorter Introduction with a clear image presenting what is done in the article.
- Has a very clear Method section, which is understandable for machine learning experts.

If the article is rewritten, I will gladly analyze it again and give more comments on the method per se. Due to the current unclearness of the article, I, unfortunately, cannot do it.

Reviewer #3 (Remarks to the Author):

Journal: nature communications

Title: Hierarchical AI enables global interpretation of culture plates in the era of digital microbiology

Number: NCOMMS-22-46898-T

In this paper, a new system (namely global interpretation of diagnostic bacterial culture plates, including presumptive pathogen identification.) is proposed to deal with the problem of bacterial culture plate analysis within a digital microbiology research background. First, a dataset is built up, where 1351 plate images, 25213 colony images and 32 microorganism species are included. Then, a two-step method framework is introduced. In the first step, single colony identification (level 0-2) is carried out. In the second step, a context-based identification is done. Finally, different evaluation values are calculated to evaluate the performance of this system. My detailed comments are as follows:

1. Application value: As mentioned in the paper, "Full Laboratory Automation is revolutionizing working habits in a steadily increasing number of clinical microbiology facilities worldwide, generating huge flows of digital images to interpret.", the application goal of this research is meaningful and would be useful for microbiology and medical data analysis.

2. Technique value: From my private feeling, I like the idea of this system very much. However, from an objective point of technology, the methodological contribution of this paper is really limited, where only some existing machine learning methods are simply modified and used to solve this special digital microbiology problem. My suggestions are below:

(1) In "Single colony identification (level 0 - 2)":

1) The proposed CNN can achieve the purpose of colony identification, however, the architecture is kind of out-of-date.

2) In Fig. 3(a), the proposed model seems like a traditional VGG-based model, which is lack of novelty.

3) Besides, the normalization (such as BN) is missed in this model, and the reason should be given.

4) The feature extraction capability of such a shallow neural network may be doubted, and the improvement of this model may improve the identification performance.

5) Maxpooling can be applied for down-sampling, but may lost many meaningful features, try to find out another down-sampling method if possible.

(2) In "Context-based Identification (level 3)":

- 1) Similarly, the Siamese CNN is also an old approach for similarity comparison of two patches.
- 2) And there is no improvement for architecture and loss function design of the proposed model.
- 3) Besides, an existing mean shift clustering is applied here, which shows few novelty value for the proposed method.

3. Data quality: In this paper, a large labeled clinical dataset is created for the training of DeepColony at the colony level. Starting from 1351 unique plate images, a dataset of 26213 isolated colony images was produced. These represent 32 UTI bacterial and fungal species, constituting 98% of the species that have been observed in three months of the clinical routine of a large CML, and are represented in their clinical variability. So, the quality of this dataset is very good.

4. Experimental result: the experimental results are good. However, there are not enough contrast experiments to show the special advantages of the proposed method. For example, the transformer-based models can be applied to compared with the proposed CNN-based models; for level 3, several traditional similarity evaluation indices such as SSIM should be applied as a contrast experiment.

5. Figure quality: The figures in this paper are not vector images and not clear to show complex contents, such as Fig. 2, 3, 5.

6. Table quality: OK.

7. Equation and mathematic quality: Some mathematical symbols are not professional. For example, in "Single colony identification", times should be "x", but not "x".

8. Language quality: The English presentation in this paper is good.

9. Reference quality: There many existing work about microorganism image analysis, but the authors did not read. I recommend the authors to read but not have to cite:

[1] Microorganism image biovolume measurement: A Comprehensive Survey with Quantitative Comparison of Image Analysis Methods for Microorganism Biovolume Measurements;

[2] Microorganism image analysis using deep learning: Applications of artificial neural networks in microorganism image analysis: a comprehensive review from conventional multilayer perceptron to popular convolutional neural network and potential visual transformer.

10. Overall evaluation: I think the application motivation and writing quality of this review paper is good, but the tech novelty and methodological contributions of the paper are very limited.

Hierarchical AI enables global interpretation of culture plates in the era of digital microbiology

Response to reviewers' remarks

Dear Reviewers, the co-authors would like to express heartfelt thanks for all the helpful and constructive comments received. We have worked to make improving changes to the manuscript, interpreting and implementing everything that has been suggested. We hope you can appreciate the work done.

Reviewer #1 (Remarks to the Author):

In this manuscript, the authors developed a hierarchical multi-network capable of handling identification, quantitation and interpretation stages, from the single colony to the whole plate level, in the challenging context of urinary tract infections.

Working on a large stream of clinical data and a complete set of 32 pathogens, the proposed system proved to be capable of effectively assisting plate interpretation with a surprising degree of accuracy. Thanks to the rich species-related generated information, the authors claim that DeepColony can be used for developing trustworthy clinical decision support services in lab automation ecosystems from the local to the global scale.

Although I do not have specific knowledge on hierarchical AI, deep learning and Convolutional Neural Networks, I read with interest this manuscript from the view point of a clinical microbiologist. As a general view, the observation that the overall accuracy was high, is very promising.

I have some general comments for the authors consideration:

- Only blood agar plates were used, on which almost all urinary bacteria are grown and the discrimination of colonies is rather difficult. Why the selective MacConkey plates were not used at all, particularly when UTIs were investigated, in which mostly Gram-negatives are yielded and the majority of Gram-positive colonies are contaminants? In MacConkey plates, other properties, such as lactose fermentation would help the identification.

First of all, thank you for appreciating the work and in particular for fully understanding the main messages in it, especially those addressed to the target readers with expertise in clinical microbiology.

The reviewer's question about the possible additional use of MacConkey agar is very pertinent and fully supportable, and of course can represent one of the most natural extensions of this work looking at the laboratory routine. The scenario in which MacConkey is used as a complementary medium is indeed common in most laboratories. However, from an identification point of view, the blood agar scenario is the most comprehensive in terms of pathogen growth, is cost-effective, and remains necessary in the context of UTIs. Furthermore, it is definitely the most challenging in terms of visual interpretation (due to the number of species that can grow and phenotypic similarity issues). We believe that demonstrating the performance of DeepColony on blood agar to the extent that we have, indicates the strength of this tool. Other solutions that work on other types of media, e.g., MacConkey and chromogens, are under investigation and can be developed to be integrated with the Blood Agar version of DeepColony to increase the sensitivity and specificity of organism identification.

- The quality of plating in figure 5 is quite low; the growth in two of the plates is rather confluent, the colony discrimination poor and the enumeration very challenging. Although the authors claim that 1- μ l loops were not used, the pictures look more of 10- μ l loop. Anyway, better discrimination and picking up of pure colonies would be needed for such a system to perform appropriately. Development of such strategy would be important for the system in the future.

- I would like to see 1-2 figures of real-life plates from positive urinary cultures (with one or two different and not so many different colony types), so that the reader may estimate the accuracy of colonies' enumeration and reporting.

We confirm that 1- μ l loop was used. The example plates in Figure 4 (formerly Figure 5) were chosen to highlight the particularly challenging picking up of single colonies in a confluent area context and the robustness of the system even in presence of highly concentrated clinical samples. In particular, for PLATE 2 the hypothetical outcome produced by skipping *level 3*, i.e., giving an interpretation of the plate at *level 4* based only on the information generated at *level 2*, would have led to discarding the plate as non-significant. Only after contextual analysis at *level 3*, the number of identified species was correctly reduced despite the unfavourable condition of high growth.

We thank the reviewer for asking more example to better appreciate and evaluate the action of DeepColony, from enumeration to reporting. This led us to show a representative collection of real-life plates that we added and described in Supplementary Fig.1, where a total of 9 plates, divided into three groups, represent different interpretive scenarios, different number of species, and various growth levels. This allowed providing a variety of examples of how the system operates and further clarified the role and importance of its multi-level structure (especially the role of *level 3*). In general, the examples demonstrate that the entire DeepColony method, in various situations, is able to regularize the number of pathogens detected without unduly reduce their actual number. In some cases this avoids losing plates that might risk being judged non-significant by simpler species identification, and in other cases it strengthens the interpretation of non-significance of plates.

- The paragraph lines 294-316 are mainly general, introductory comments and not methods.

We are in full agreement. The paragraph was removed from the methods, condensed, and inserted into the introduction to make it clearer.

- It would be of considerable interest if other types of clinical samples, further than UTI ones, were also tested.

We too hope that this study will inspire others on different types of clinical samples. Blood agar is also used in other clinical settings, and again, this makes DeepColony a good basis for further work.

Reviewer #2 (Remarks to the Author):

I want to start this review with a clear statement that I have a pure machine learning background with a few papers in automatic microbiology. Moreover, I assume that Nature Communication is not a purely microbiological journal, like Trends in Microbiology, and articles presented in this journal should be directed to a wider audience than only microbiological experts. Therefore, my review should be considered as a review from a potential Nature Communication reader with a machine learning background.

Overall, it is clear to me that the authors have extensive knowledge in microbiology and have published many papers on this topic. However, the article is very unclear for somebody who does not have such an

experience. The descriptions are too long and very complicated, and I guess only a few people in the world who are on the same level of expertise as the authors would be able to read this article and get something interesting from it.

We thank the reviewer for his clarifying background statement. We confirm that this work is deeply interdisciplinary, and we acknowledge that readers without a microbiology background may have some problems in fully understanding the importance and impact of some key aspects. We have seriously considered the perspective of both main types of readers' backgrounds (microbiology and machine learning) in all our revisions, particularly by curating consistency of the logical flow in the introduction and methods, improving the key figures, and adding supplementary material and figures.

Taking this into consideration, I decided to read Intro, Results, and Discussion very fastly and concentrate on the Method section, which I thought should be of interest to somebody like me - i.e. machine learning expert who cooperate with microbiologists.

Unfortunately, the Method section was also unclear to me. Figure 1a (Overview of the DeepColony system) brings more confusion than clarifications. There is not a single figure of network architecture.

Thank you for explicitly pointing out issues when reading the Methods before or after a quick reading of the rest of the paper. This helped us greatly to improve the readability of the document. Anything that interrupted the logical flow in the description of the methods has been removed/revised/relocated. We added a supplementary material file with notes, tables and figures on various aspects related to methods, data and results (more details are given in the various specific responses).

We have made significant updates to Figure 1, following the reviewer's suggestions. First, we added new subfigure 1a, showing the overall workflow, to effectively illustrate where our contribution is located within the FLA environment. Secondly, we have graphically redesigned the hierarchical structure of the multi-network processing/analysis architecture (new subfigure 1b), giving greater clarity through the use of exemplifications of intermediate results (depicted in subfigures 1c-1e). In addition, we have included two new subfigures 1f and 1g, which represent the convolutional networks involved in *levels 2 and 3* of the hierarchical architecture. These updated subfigures provide a more refined and accurate representation of the network architectures than the previous versions presented in Figure 3. It is important to note that DeepColony is not based on a single deep neural network architecture, but features multiple networks working collaboratively according to a hierarchy of subtasks, which map both procedurally and cognitively to the task of interpreting clinical plates.

Reading the paragraph "Single colony identification", I first thought that the authors use simple CNN architecture, but then "Context-based Identification" appeared with "Siamese" networks, and the description is too fuzzy to understand how they are used and why?

The two CNN and Siamese networks work in cascade as they support a cognitive flow that governs a particularly complex visual identification task. This task involves focusing on individual viable colonies and then adjusting the identification assumptions based on an overview of the entire plate. In rewriting the Methods section, a paragraph was inserted at the beginning of the processing part to prevent disorienting the readers, especially those who prefer to look at the methods first. In it, we introduce (or recall) the hierarchical structure of the multi-network architecture, making explicit reference to the new Fig.1 and thus creating the framework for interpreting the individual paragraphs that follow.

The authors bring too many implementation details, like PreReLU, momentum, etc, which should be moved to supplementary materials.

Thank you for the suggestion. We rewrote almost the entire description of the methods to make them clearer and moved detailed aspects to the supplementary materials. To lighten the description of the

methods from too many technical details and to increase self-consistency, we also worked on a better graphical representation of the details of the identification networks (single colony and context-based) in subfigures 1f and 1g.

In my opinion, the current version of the article should not be accepted, and the editor should ask the authors to rewrite the whole method section using best practices from other articles already published in Nature Communication, like this one: <https://www.nature.com/articles/s41467-019-12898-9>

This already-published article, in contrast to the reviewed one:

- *Has a shorter Introduction with a clear image presenting what is done in the article.*
- *Has a very clear Method section, which is understandable for machine learning experts.*

If the article is rewritten, I will gladly analyze it again and give more comments on the method per se. Due to the current unclarity of the article, I, unfortunately, cannot do it.

The suggested paper has been taken as a reference. However, differently from a single network approach, we have an articulated processing/analysis system that requires the cooperation of multiple networks on a hierarchy of tasks, and this inevitably adds a level of complexity that cannot be reduced. We hope to have better managed these critical aspects thanks to the reviewer's suggestions. We also cited the paper in the introduction (with the other promising/reference non-culture techniques) because it represents a good reference in the context of identifying bacteria with vibrational spectroscopy techniques.

Reviewer #3 (Remarks to the Author):

In this paper, a new system (namely global interpretation of diagnostic bacterial culture plates, including presumptive pathogen identification.) is proposed to deal with the problem of bacterial culture plate analysis within a digital microbiology research background. First, a dataset is built up, where 1351 plate images, 25213 colony images and 32 microorganism species are included. Then, a two-step method framework is introduced. In the first step, single colony identification (level 0-2) is carried out. In the second step, a context-based identification is done. Finally, different evaluation values are calculated to evaluate the performance of this system. My detailed comments are as follows:

1. Application value: As mentioned in the paper, "Full Laboratory Automation is revolutionizing working habits in a steadily increasing number of clinical microbiology facilities worldwide, generating huge flows of digital images to interpret.", the application goal of this research is meaningful and would be useful for microbiology and medical data analysis.

2. Technique value: From my private feeling, I like the idea of this system very much. However, from an objective point of technology, the methodological contribution of this paper is really limited, where only some existing machine learning methods are simply modified and used to solve this special digital microbiology problem.

We thank the reviewer for highlighting the valuable aspects of this work and for the constructive criticisms, which helped us to improve the work in the direction of better clarifying the technological choices and novelty aspects. Regarding the latter aspect, we can reason on two levels: 1) the technical novelty lies not in the individual networks but in the hierarchical multi-network and multilevel architecture that allows mapping a complex multiparametric decision-making process oriented towards the clinical interpretation of the entire culture plate; 2) there are precise reasons for the choice of individual network architectures that we recognize were not justified clearly enough in the first version of the manuscript. In the revision we better justify the relative simplicity of the individual networks by linking it to the particular nature of the data and tasks involved, highlighting aspects of overall consistency of the architecture and operating targeted tests, also in response to the various specific suggestions received.

We also emphasize the novelty of the particular multi-network orchestration we propose, and in particular the role of level 3 in regularizing single colony species identification to produce a reliable and safe global plate interpretation. This is also further demonstrated by the various clinical cases presented in Supplementary Figure 1.

We avoided introducing alternative technology choices that might be too much of a substitute for the microbiologist's work or fail to provide the necessary level of transparency and explainability. We chose to work on a system architecture able to support the microbiologist's work, with solutions that could also generate value-added information and promote a high versatility of use in different laboratory automation contexts.

Overall, we appreciate the reviewer's feedback and feel that our work has been strengthened by their suggestions.

My suggestions are below:

(1) In "Single colony identification (level 0 - 2)":

1) The proposed CNN can achieve the purpose of colony identification, however, the architecture is kind of out-of-date.

2) In Fig. 3(a), the proposed model seems like a traditional VGG-based model, which is lack of novelty.

As anticipated above, the novelty aspects are to be found in the multilevel multi-network visual analysis architecture, which succeeds in mapping a complex cognitive process. Moreover, the various networks targeted on specific visual tasks were all selected in their best configuration, following careful design and sizing processes. Interestingly, the resulting configurations are highly consistent with each other despite the diversity of subtasks. Such CNN architectures while consolidated and relatively simple are also ideal for keeping processing time and server resources under control. This matches favourably with side objectives related to real-time operation needs in FLA environments. In particular, many more advanced solutions have been tested that, in addition to not introducing performance improvements, involve higher computational loads that can become critical for intensive workloads and resource-constrained environments. The main dimensioning/ablation tests we performed have been reported in the supplementary materials and further commented on below in response to the other specific comments.

3) Besides, the normalization (such as BN) is missed in this model, and the reason should be given.

We are sorry to have created a misunderstanding. Normalization is actually done and is described in the methods. In the case presented, it is implemented in the form of Local Response Normalization, LRN. It was not even a well explained detail in the figures, whereas now it also appears in the visual block legend of the new subfigures 1.f and 1.g.

4) The feature extraction capability of such a shallow neural network may be doubted, and the improvement of this model may improve the identification performance.

This is a very interesting point and gives us the possibility to go a little deeper beyond the experimental evidence (with apologies for the length of this answer). On the one hand, we are dealing with rather simple objects (bacterial colonies) that can be conceived of as having less "dimensions of variability" (despite clinically significant) when compared to other object classes, such as cats or dogs. This contributes to justify why our system dimensioning leads to CNN networks to be not highly deep, but still able to effectively capture and represent the variabilities within this object category. On the other hand, more than and

differently from many other common object recognition/classification problems, very subtle differences can be highly significant for the correct identification of different classes (pathogens) that are known to possibly exhibit a high level of similarity. This requires different spectacles in interpreting the correct matching between problem and tools than the usual capacity/novelty criteria. This is the strength of our incremental approach that follows (and confirms!) the clinical decision-making flow: from quantification to the selection of good reference colonies, to the first presumptive colony-oriented identification and, in the next contextual phase, to the reconsideration of the identification according to ensemble similarity criteria, where (as further specified below) similarity is an articulated concept here, going beyond direct visual similarity. For these reasons, it is not so much the capacity/complexity of the model that matters, but a correct expressiveness ability combined with a good adherence to the cognitive assessment and decision-making phases. This is also why increased model complexity did not experimentally lead to performance improvement. Clear evidence of the minimal effect of higher complexity models in our multinet case is shown in the new Supplementary Table 2. Furthermore, higher complexity (e.g. the use of a visual transformer making whole plate significance decisions) is also critical considering the need of a transparent decision-making process and good model explainability, the need to easily adapt to different sets of laboratory rules (without retraining on each dataset fragment), as well as the need for a controlled computational complexity (so as not to introduce feasibility/cost barriers).

5) Maxpooling can be applied for down-sampling, but may lost many meaningful features, try to find out another down-sampling method if possible.

We conducted ablation studies to justify the optimality in this case of Maxpooling, aware of some existing criticisms of this downsampling strategy and alternative approaches. However, in addition to being a computationally efficient solution, we experienced that in our case Maxpooling is better than other popular alternatives. The results of comparisons with average pooling and strided convolutions are reported in Supplementary Table 2. The superiority of Maxpooling can be intuitively justified because our context offers the ideal application of ideas behind the proposition of Maxpooling as a solution for downsampling, i.e. (partial) compensation of shift variance introduced by subsampling processes. In fact, in our case of images coming from the single colony dataset, i.e. representing round colonies at the centre of the image patch, maxpooling allows the desired features to be captured more consistently, regardless possible small shifts of the colony centre, of the borders and other distinctive traits or possible small geometric distortions. Again, therefore, a key role is played by the fact that colonies are not just any object and that the high level of culture standardization than can be ensured by laboratory automation (which allows for stable and repeatable culturing conditions) allows the network to be expected to be able to mostly highlight and capture the variations that really may be associated with species differences.

(2) In "Context-based Identification (level 3)":

1) Similarly, the Siamese CNN is also an old approach for similarity comparison of two patches.

2) And there is no improvement for architecture and loss function design of the proposed model.

3) Besides, an existing mean shift clustering is applied here, which shows few novelty value for the proposed method.

In addition to the subsistence of the previous considerations in terms of complexity, expressiveness, and overall consistency within the multi-network architecture, we have found that the Siamese approach is extremely effective for our specific application, while maintaining minimal computational impact, making it a practical solution for real-world scenarios. The originality of the approach lies in the fact that we do not simply ask the network to tell whether two colonies are similar or not. Here the Siamese CNN is used in a less conventional way to perform dimensionality reduction embedding that is subsequently exploited to infer contextualized similarity at the plate level through a flexible clustering strategy. Thus, beyond the tools taken individually, it is their combination conceived as clustering in an embedding space that gave us

a real opportunity to smoothly revise previous identification outcomes and that proved to be a powerful tool to improve plate interpretation performance by a significant margin.

3. Data quality: In this paper, a large labeled clinical dataset is created for the training of DeepColony at the colony level. Starting from 1351 unique plate images, a dataset of 26213 isolated colony images was produced. These represent 32 UTI bacterial and fungal species, constituting 98% of the species that have been observed in three months of the clinical routine of a large CML, and are represented in their clinical variability. So, the quality of this dataset is very good.

Thank you we only confirm that the datasets have been judged as highly representative of the real-world and actual needs.

4. Experimental result: the experimental results are good. However, there are not enough contrast experiments to show the special advantages of the proposed method. For example, the transformer-based models can be applied to compared with the proposed CNN-based models; for level 3, several traditional similarity evaluation indices such as SSIM should be applied as a contrast experiment.

Thank you for these points. For what concerns the considerations and experiments to demonstrate the advantages of the proposed solutions please refer to the above answers (especially the one related to suggestion (1).4).

In terms of possible comparison with more traditional similarity evaluation indices, such as the structural similarity index SSIM, we have done an experiment (see new Supplementary Figure 2) in which, while the Siamese CNN + clustering guarantee 99.1% accuracy, discrimination based on SSIM thresholding never goes over 83% accuracy, with drastic penalty for SSIM values over 0.7. This is a clear evidence that we are not seeking for simple visual similarity, but for a higher conceptual level of similarity, which must encompass and accommodate for different colony sizes (due to different nourishment conditions over the plate) and, most importantly, intra-species polymorphism. In other words, our dataset has many similar colonies belonging to different strains, and many different colonies belonging to the same strain, so there is the need of a (Siamese) learning procedure rather than an objective metric. Visual examples of Inter-species similarity vs Intra-species differences have been reported in a new Supplementary Figure 3.

5. Figure quality: The figures in this paper are not vector images an not clear to show complex contents, such as Fig. 2, 3, 5.

Some conversion/degradation occurred during the submission and we will verify this with the editors. Since colony/plate images are high resolution and some overlays are natively raster based, we will do our best to preserve the quality of the images and/or to identify a way to access the images in their full resolution (on our side the quality is perfect, therefore there should be a solution to the problem).

6. Table quality: OK.

7. Equation and mathematic quality: Some mathematical symbols are not professional. For example, in "Single colony identification", times should

be "x", but not "x".

Thank you. Fixed in both main text and captions.

8. Language quality: The English presentation in this paper is good.

9. Reference quality: There many existing work about microorganism image analysis, but the authors did not read. I recommend the authors to read but not have to cite:

[1] *Microorganism image biovolume measurement: A Comprehensive Survey with Quantitative Comparison of Image Analysis Methods for Microorganism Biovolume Measurements*

[2] *Microorganism image analysis using deep learning: Applications of artificial neural networks in microorganism image analysis: a comprehensive review from conventional multilayer perceptron to popular convolutional neural network and potential visual transformer.*

Thank you for bringing these recent articles to our attention. Actually, the second one gave us an updated confirmation that so far in the field of MIA (microbiology image analysis) nothing has been proposed that comes close to solving the clinical interpretation of culture plates in its full complexity. We cited the paper in the introduction among others that confirm this evidence.

10. Overall evaluation: I think the application motivation and writing quality of this review paper is good, but the tech novelty and methodological contributions of the paper are very limited.

We are deeply grateful for the opportunity we have had to address concerns regarding novelty aspects of our work. Through various clarifications, experimental trials and supplementary material to our manuscript, we have sought to ensure that these aspects are now more robust and well integrated with the other highlights of our work, particularly in terms of potential applications and impact.

REVIEWER COMMENTS

Reviewer #1 (Remarks to the Author):

My comments were addressed adequately.

Although the manuscript is a bit complicated and not very applicable in the clinical practice, the idea is valid as a starting point and worths publication.

Reviewer #2 (Remarks to the Author):

Firstly, I want to thank the authors for their thoughtful responses to my review. I think they put a lot of work into preparing the new version of the manuscript. Unfortunately, the current version is still far from perfect due to the following reasons:

- Paragraph "DeepColony CNN architectures and training" should contain information about the overall pipeline with references to detailed descriptions in the following paragraphs. In fact, this paragraph should reference some pseudo-code that describes the pipeline more formally. Now, it is still unclear what are the successive steps and how they depend on each other...
- Paragraphs after "DeepColony CNN architectures and training" still contain too many details about hyperparameters etc. It should be moved to the supplementary materials. This way, paragraphs like "Single colony identification" should limit to one sentence about using CNN architecture to identify a single colony.
- More effort should be put into describing the pipeline because, as I understand, it is the main achievement of the paper. Using CNN is no novelty whatsoever.
- Considering the paragraph "Nonlinear similarity-driven embedding", I still have no idea why the authors use Siamese networks. Maybe it will clarify by pseudo-code.
- When it comes to Fig. 1 (the most important figure of the paper), I really enjoy parts a, f, and g. However, parts b-e are still unclear. They should be moved to separate figures and redrawn.
- First three paragraphs from section Methods should be in the other section called, e.g., Dataset.

Reviewer #3 (Remarks to the Author):

After checking the updated paper, I found that all my comments were solved and the quality of this paper had been improved a lot to close to a final publication.

Hierarchical AI enables global interpretation of culture plates in the era of digital microbiology

Response to reviewers' remarks

Dear reviewers, all co-authors would once again like to express their heartfelt thanks for your work and all the helpful and constructive suggestions that enabled us to significantly improve our article. We hope you will appreciate the work done.

Reviewer #1 (Remarks to the Author):

My comments were addressed adequately.

Although the manuscript is a bit complicated and not very applicable in the clinical practice, the idea is valid as a starting point and worths publication.

First of all, thank you for appreciating our revision work. We fully understand your comment on the importance of application value and assure you that this has always been our priority as well. The results of this study demonstrate that a multi-network deep-learning approach is able to accurately enumerate and classify bacterial colonies in routine culture. This indeed is a good starting point to concretely develop a clinical decision support tool for the clinical microbiology laboratory in determining how to properly work up a specific culture. Given the highly manual processes and large labor shortage in the clinical microbiology laboratory, a tool such as this would have tremendous impact on the overall workflow in the laboratory. Additionally, this work wants to provide the foundation to develop future applications for other colony morphologies and culture plates to be expanded to cover the entire breadth of clinical microbiology.

Reviewer #2 (Remarks to the Author):

Firstly, I want to thank the authors for their thoughtful responses to my review. I think they put a lot of work into preparing the new version of the manuscript. Unfortunately, the current version is still far from perfect due to the following reasons:

- Paragraph "DeepColony CNN architectures and training" should contain information about the overall pipeline with references to detailed descriptions in the following paragraphs. In fact, this paragraph should reference some pseudo-code that describes the pipeline more formally. Now, it is still unclear what are the successive steps and how they depend on each other...

We are very grateful to the reviewer for his guidance in improving the clarity of our work. We very much welcomed the suggestion to introduce a more in-depth section about the architecture, as well as those to make Figure 1 even more usable, both for data analysis experts and microbiologists, and that of introducing a pseudocode of the entire architecture. We are now hopeful that the work will indeed lead in the direction indicated by the reviewer. In the Methods we turned, as suggested, what was only an introduction into a section "DeepColony architecture" with complete information about the overall pipeline with references to detailed descriptions of the core parts in the following paragraphs. In the paragraph we reference a complete and detailed pseudo-code, included in the supplementary material, that describes the pipeline formally.

- Paragraphs after "DeepColony CNN architectures and training" still contain too many details about hyperparameters etc. It should be moved to the supplementary materials. This way, paragraphs like "Single colony identification" should limit to one sentence about using CNN architecture to identify a single colony.

Thank you also for this suggestion that we implemented. Now only essential info about the network structure and training remain in the Methods, while details have been integrated into the Supplementary material. This and the other suggested changes make the Method section more interesting for both computer scientists and microbiologists.

- More effort should be put into describing the pipeline because, as I understand, it is the main achievement of the paper. Using CNN is no novelty whatsoever.

Yes, as we have already acknowledged, the choice of technologies was determined by the type of data and the nature of the subtasks, but the main contribution is precisely the overall architecture and results on complex tasks that have not been addressed with these approaches so far. All the effort in this second revision was done in the suggested direction.

- Considering the paragraph "Nonlinear similarity-driven embedding", I still have no idea why the authors use Siamese networks. Maybe it will clarify by pseudo-code.

Thank you for pointing out this, and sorry for not having captured it in the previous revision. We realized that we mistakenly took for granted a detail about how the Siamese network is used. After a training of the Siamese-CNN directed to capture an inter-strain "similarity", encompassing possible polymorphism and contextualized on what can happen on the single plate, in runtime we use one of the twin branch to obtain a feature vector that embed the single colony image in a similarity space where we expect colonies from the same strain (despite possible polymorphism expressed on the plate) to be close so that a clustering can be meaningfully used to guide a regularization of previously estimated species-ID rankings. This is very close to the visual and cognitive "regularization" reasoning the microbiologist actually does while interpreting the growth on a specific plate. All this is now clarified in the Methods (both in the "DeepColony architecture" and in the "Nonlinear similarity-driven embedding" paragraphs) and in the Supplementary pseudocode.

- When it comes to Fig. 1 (the most important figure of the paper), I really enjoy parts a, f, and g. However, parts b-e are still unclear. They should be moved to separate figures and redrawn.

Thank you for the suggestion. We kept exactly the indication even if operate slightly differently. In order not to deplete too much Figure 1, we simplified the Figure 1b making it more self-explanatory (the role of the various levels are now reported directly on the blocks) and highlighting the main modules of the architecture, however proposing a linear flow and deferring the complexity of block links and dependencies to the explanations and to the pseudo-code. Figure 1c-1e have a great value for immediate understanding by microbiologists of the action of the various levels and of the proposed contribution. A better explanation of these subfigures is now present in the new "DeepColony architecture" section. They are now also better directly interpretable given the more self-explaining nature of Fig.1b, therefore they have not been removed/replaced.

- First three paragraphs from section Methods should be in the other section called, e.g., Dataset.

We named, as suggested, a section "Datasets" comprising all descriptions about data.

Reviewer #3 (Remarks to the Author):

After checking the updated paper, I found that all my comments were solved and the quality of this paper had been improved a lot to close to a final publication.

Sincere thanks for all the opportunities we received from the reviewer that allowed us to improve our work.

REVIEWERS' COMMENTS

Reviewer #2 (Remarks to the Author):

Figure 1 looks really clear and impressive. I think it will be great figure to show in any presentations that the authors will give about the paper. And, I am glad I could help with improving it.
Paragraph DeepColony architecture also improved a lot. I like the overall description at the beginning and details limited to the most important things.
Pseudo code also looks fine.

One minor thing:

I propose to start each „At level...” in DeepColony architecture paragram from new line for clarity.

I recommend to accept the paper.

Nature Communications manuscript submission: NCOMMS-22-46898-B

Hierarchical AI enables global interpretation of culture plates in the era of digital microbiology

Response to reviewers' remarks

Reviewer #2 (Remarks to the Author):

Figure 1 looks really clear and impressive. I think it will be great figure to show in any presentations that the authors will give about the paper. And, I am glad I could help with improving it.

Paragraph DeepColony architecture also improved a lot. I like the overall description at the beginning and details limited to the most important things.

Pseudo code also looks fine.

We are grateful for the valuable suggestions we received, we are pleased to see that we were able to interpret them correctly, and we are also satisfied with the end result.

One minor thing:

I propose to start each „At level...” in DeepColony architecture paragram from new line for clarity.

Done

I recommend to accept the paper.